# Manufacture and Characterization of Polylactic Acid Filaments Recycled from Real Waste for 3D Printing

**DOI:** 10.3390/polym15092165

**Published:** 2023-05-01

**Authors:** Saltanat Bergaliyeva, David L. Sales, Francisco J. Delgado, Saltanat Bolegenova, Sergio I. Molina

**Affiliations:** 1Department of Materials Science and Metallurgical Engineering and Inorganic Chemistry, Algeciras School of Engineering and Technology, Universidad de Cádiz, INNANOMAT, IMEYMAT, Ramón Puyol Ave, 11202 Algeciras, Cádiz, Spain; david.sales@uca.es; 2Physics and Technology Department, Al-Farabi Kazakh National University, 71, Al-Farabi Ave, Almaty 050040, Kazakhstan; saltanat.bolegenova@kaznu.edu.kz; 3Department of Materials Science and Metallurgical Engineering and Inorganic Chemistry, Universidad de Cádiz, Campus Río S. Pedro, INNANOMAT, IMEYMAT, 11510 Puerto Real, Cádiz, Spain; fjavier.delgado@uca.es (F.J.D.); sergio.molina@uca.es (S.I.M.)

**Keywords:** polylactic acid, recycling, waste, additive manufacturing, fused filament fabrication, thermogravimetry, calorimetry, scanning electron microscopy, tensile strength

## Abstract

This paper studies the thermal, morphological, and mechanical properties of 3D-printed polylactic acid (PLA) blends of virgin and recycled material in the following proportions: 100/0, 25/75, 50/50, and 75/25, respectively. Real waste, used as recycled content, was shredded and sorted by size without a washing step. Regular dog-bone specimens were 3D printed from filaments, manufactured in a single screw extruder. Thermogravimetric analysis indicated that adding PLA debris to raw material did not significantly impact the thermal stability of the 3D-printed samples and showed that virgin and recycled PLA degraded at almost the same temperature. Differential scanning calorimetry revealed a significant reduction in crystallinity with increasing recycled content. Scanning electron microscopy showed a more homogenous structure for specimens from 100% pure PLA, as well as a more heterogeneous one for PLA blends. The tensile strength of the PLA blends increased by adding more recycled material, from 44.20 ± 2.18 MPa for primary PLA to 52.61 ± 2.28 MPa for the blend with the highest secondary PLA content. However, this study suggests that the mechanical properties of the reprocessed parts and their basic association are unique compared with those made up of virgin material.

## 1. Introduction

Additive manufacturing (AM) technologies, also called 3D printing or rapid manufacturing, are among the enabling technologies in Industry 4.0. These technologies are associated with potentially strong stimuli for sustainable development to time- and cost-saving [1,2,3,4]. The most widely used AM process is fused filament fabrication (FFF) due to its simplicity, low running, and material costs [5]. In addition, FFF prices can go as low as 0.20 €/part, thus making it an attractive technology [6]. Wittbrodt et al. [7] showed that even making the extremely conservative assumption that a household would only use the 3D printer to make a selected 20 products a year, the avoided purchase cost savings would range from about 300 to 2000 USD/year.

Despite these benefits, 3D printing still creates a significant amount of waste [8]. Filament material is widely thrown out during the fabrication process due to printing failures, broken parts, filament replacement, discarded support structures, and nozzle tests [5]. Moreover, some printed products are used as prototypes that could be discarded at the end of the product development process [9,10]. Hence, a certain increase in the amount of plastic waste associated with the development of the polymer AM market could be predicted. However, previous studies have shown that over 40% of material-related waste could be avoided using AM, as well as 95% of unused material could be reused [11]. Costs could also be reduced by recycling locally, where individuals, groups, or small businesses could procure recycling equipment in the range of 3000 USD [8].

Polylactic acid (PLA) is among the most popular polymers in FFF. It is a biodegradable and renewable thermoplastic polyester derived from renewable sources (mainly starch and sugar). It replaces conventional petrochemical-based polymers, such as acrylonitrile butadiene styrene, and reduces oil consumption by 30–50% [12,13,14,15]. The global production of PLA was around 0.2 million tons in 2017 [16] and around 0.19 million tons in 2019 [17]. In addition, the annual output of PLA is expected to reach 0.56 million tons by 2025, an increase of 53.8% over 2020 [18]. The ratio between the price of virgin PLA and the price of PLA recyclate from post-industrial waste is 46% and 51% for post-consumer [19].

PLA is supposed to finish its life in compost [12]. It degrades slowly in nature, taking between 2 and 10 months in dry conditions [20]. Because of this degradation stability of PLA products in soils at ambient temperatures, there is still the risk of environmental contamination [19]. However, recycling PLA filaments for 3D printing is a feasible option, as it offers environmental benefits, such as reducing landfill and CO_2_ emissions from waste transportation [21]. In this sense, a comparative life cycle assessment conducted on meat trays showed that PLA production (in the form of granulates) contributes 63% to its overall greenhouse gas emissions, with this rate being greater than that for amorphous polyethylene terephthalate (PET, 53%) and polypropylene (PP, 44%) [22]. Additionally, over 80% of the embodied energy used in transportation and collection could be saved [23], distributed recycling and manufacturing methods could reduce energy consumption by a factor of two compared with traditional manufacturing ones [24], and making 3D printing filament at home from recycled filament saves about 40 times more energy than commercial production [25].

Previous works have shown that 3D printing is a cost-effective recycling process that economizes natural resources and time. However, the physical and chemical properties of recycled plastics should be studied to make them effective. Zander et al. showed that FFF filament from 100% recycled bottles and packaging PET without any chemical modifications or additives is a viable new feedstock for FFF, with the mechanical properties of the printed parts comparable to the parts made from commercial filament [26]. In work [27], Zander et al. processed blends of PET, PP, and polystyrene waste into filaments for 3D printing with tensile strengths comparable to some lower-end common commercial filaments, such as high-impact polystyrene. Fabio et al. recycled PLA in an open-source additive manufacturing context [28]. As shown in [29], low-density polyethylene (LDPE)/linear LDPE is a soft, low-modulus, high-toughness polymer, which leads to a variety of additive manufacturing complications. In [25,30], filament made of recycled high-density polyethylene (HDPE) pellets had favourable water rejection, with an extrusion rate and thermal stability comparable to those of the filament made of acrylonitrile butadiene styrene pellets. According to data from studies, polymers could only be recycled to a lower material level (downcycling), so they could be used in less critical applications because of worse mechanical properties [8,9,10,28]. To improve the quality of recycled polymers, composites were created from polymer waste by adding natural/synthesis particles and/or fibres of a micro- or nanoscale size. For instance, in [31,32], the authors manufactured 3D printable composites based on the waste of PP by adding glass fibre in the first work and basalt fibre in the second one. The produced filament with 5% wt. of basalt had the highest tensile strength among all the produced samples [32]. It must be mentioned that the quality and processability of fabricated filaments mainly depend on extrusion parameters, such as the rpm/speed of the screw, extrusion temperature, and load. These extruder settings are optimized experimentally and can vary. According to the above-mentioned studies, PET filament was extruded at a temperature that is equal to the melting point. To fabricate filament from PP, LDPE, and HDPE, the extrusion temperature was set up much higher than the melting temperature. The opposite is true for PLA.

Manufacturing materials mainly affect the quality and performance of parts made of FFF. The mechanical effects of recycled PLA content on PLA filaments should, therefore, be studied to attain the desired quality characteristics in the parts developed by the FFF process. Studying the effect of the parameters on the response characteristics of the FFF parts helps to adjust the level of the process variable that improves the parts’ quality [33].

In this paper, filaments from 100% virgin PLA commercial pellets were blended with real FFF-printed PLA waste in three different proportions (25, 50, and 75 weight percentages) to produce 1.75 mm diameter filaments for 3D printing. A tensile test, differential scanning calorimetry (DSC), and thermogravimetric analysis (TGA) were used to study the properties and thermal stability of the 3D-printed parts from the produced filaments. Scanning electron microscopy (SEM) was also carried out to observe the structure of the PLA blends. To the best of our knowledge, there is no previous report about filament fabrication and properties studies on 3D-printed samples prepared from neat and real PLA waste from AM. In previous studies, reprocessed polymers, not used in real-life conditions, were used as the recycled content in the blend. Therefore, this experiment offers a realistic approach to the evaluation of PLA recyclability.

## 2. Materials and Methods

### 2.1. Materials

Commercially available PLA pellets, Smartfil^®^PLA3D850, from SmartMaterials 3D (Jaen, Spain) and with a 1240 kg/m^3^ density and printing temperature of 210 ± 10 °C [34] were used as primary PLA. Figure 1 shows the first steps to making PLA pellets from 3D-printed post-consumer plastic and manufacturing the filament from them. Real debris was collected for about one year, so PLA waste had a different composition of PLA plastics with dissimilar life cycles. This age of the plastic waste was chosen because a previous study [35] revealed that 3D-printed laboratory accelerated-aged PLA samples have the same thermo-mechanical properties before reaching 1.5 years of age. Therefore, they could be recycled together.

The recycled fraction of PLA was prepared by shredding PLA waste using a Retsch SM300 (Dusseldorf, Germany). Afterward, it was sorted according to particle size using an electric sieve (Fritsch, Idar-Oberstein, Germany) with square section holes of 5, 2.5, 1.25, 0.63, 0.32, 0.16, and 0.080 mm. The particle size distribution showed 65% of the sieved PLA in the 1.25 mm fraction, so this fraction was used for filament production. 

### 2.2. Filament Fabrication

Likewise, both virgin and recycled PLA were dried in a vacuum oven, VACUtherm VT 6025 from Thermo Scientific (Waltham, MA, USA), at 50 °C overnight before extrusion to avoid the hydrolysing effect of absorbed water. Different proportions of virgin and recycled PLA pellets (Table 1) were introduced in a laboratory-sized, co-rotating, single-screw extruder, Noztek Pro Filament Extruder (Shoreham, West Sussex, UK), with one controllable heating zone for melt mixing and filament extrusion. 

The temperature of the mixing zone in the extruder was 225 °C to produce V100R0 and V75R25 filaments, and it was reduced to 205 °C to produce V50R50 and V25R75 filaments. The temperature was changed to process V50R50 and V25R75 blends better. During filament manufacturing, the produced filament had high fluidity to form a filament with a constant diameter. The temperature was reduced as regards the greater recycled PLA content due to the chain scission in post-consumer polymer structures. The Noztek extrudes to a speed of 40 rpm. Likewise, a fan was placed near the extrusion nozzle to rapidly cool the material when it came out of the nozzle. This was turned on to ensure that the filament diameter remained as close as possible to the desired diameter of 1.75 mm. The produced filament showed a constant diameter, and its surface finish was comparable to the virgin PLA filament. About 30 m of filament was produced from 0.1 kg of the PLA blends, with an average diameter of 1.7 mm. There was no filament from 100% PLA waste because the produced filament became curly (the last picture in Figure 1), so there was no printable quality.

### 2.3. Samples 3D Printing

Regular dog-bone (type 1BA) specimens from each PLA blend were printed considering the ISO 20753 standard [36] and using an Anycubic Kossel Delta Rostock Die kit 3D printer (Shenzhen, China), with a platform temperature of 60 °C. All specimens were printed with a 100% infill, horizontal pattern orientation of 0/90 (i.e., alternating layers with orientations at 0° and 90°), and a layer height of 0.2 mm. These deposition pattern orientations, together with the layer height, provided the overall maximum values for PLA compared with a 45/45 orientation [37,38,39,40,41].

### 2.4. Samples Testing

After 3D printing, test specimens were conditioned before testing them for more than 88 h at 23 ± 1 °C air temperature and 50 ± 5% relative humidity, according to ISO 291 [42]. All experiments were conducted in a standard atmosphere.

TGA was performed to identify the temperature at which the material started to chemically degrade. Approximately 15 mg of the polymer was heated at 25 °C and equilibrated for 15.00 min. Afterward, samples were heated at a 10.00 °C/min rate up to 430.00 °C under 10 mL/min of N_2_ flow. The temperature of 5% wt. mass loss (*T*_*loss*5%_) and initial and final degradation temperatures (*T_i_* and *T_f_*) were determined from the received TGA curves and according to ISO11358-1 [43].

DSC measurements give information about the structural changes of the polymer during the thermo-mechanical recycling process. DSC was carried out with a thermo-mechanical analyser, Q20, from TA Instruments Inc. (New Castle, DE, USA). Specimens weighing 5–10 mg were heated up at 10 °C/min and cooled at 2 °C/min under a nitrogen flow of 10 mL/min. The thermal properties of the specimens, i.e., the glass transition temperature (*T_g_*), crystalline melting point (*T_m_*), and crystallization point (*T_c_*), were determined. The enthalpy of crystallization and fusion (Δ*H_c_* and Δ*H_m_*, respectively) were counted using TA Universal Analysis software from TA Instruments Inc. (New Castle, DE, USA).

SEM was used to observe the microstructure of the PLA blends with a Fei Nova Nanosem (Waltham, MA, USA). Before measurements, the broken parts of the specimens were coated with gold to avoid electron charging.

Finally, the tensile test was performed using a universal testing machine, the Shimadzu AG-X series (Kyoto, Japan). Five dog-bone 3D-printed specimens from every PLA blend were tested. The width and thickness were measured using a digital micrometre, with an accuracy of 0.01 mm at multiple points [44]. The averaged mean of three cross-section measurements was taken as the measurement result. Speed was at a constant rate of 1 mm/min. During this procedure, both the load sustained by the specimens and elongation were measured. Likewise, an extensometer with a nominal length of 20 mm was used. The results of the tensile test for the 3D-printed materials were transmitted using Trapezium software version 1.5.1. Tensile strength and ductility were also calculated [44].

## 3. Results and Discussion

### 3.1. Thermogravimetric Analysis

The averaged thermograms, depicting the evolution of the weight versus the temperature of the 3D-printed specimens, are shown in Figure 2a. All the samples had a curve shape with single-stage mass reduction. In addition, V100R0 sharply dropped in mass at the end of the graph (320–350 °C section), unlike the recycled PLA blends. As the percentage of PLA waste increased, the slope of this part of the graph became more gradual. The reason could be the presence of dust and/or contaminants that could occasionally be mixed in the blends during the filament preparation process [45]. Another reason could be that the PLA filament for FFF has special additives and/or fillers used by manufacturers to improve the 3D-printed parts’ quality, so their degradation could influence the decomposition of the PLA blend. This degradation behaviour could, therefore, be the result of various chain end structures, which initiate different degradation reactions [46].

Table 2 summarizes the TGA data analysis of the PLA blends: *T_5%loss_*, *T_i_*, *T_f_*, the difference between *T_f_* and *T_i_* (*T_f_-T_i_*), and the maximum temperature of polymer degradation (*T_max_*) computed according to ISO 11358-1:2014 [43]. As a result, the 5% mass of the recycled blends was reduced at a slightly lower temperature compared with V100R0; therefore, adding debris did not significantly influence the thermal stability of the PLA blends. The accurate temperature at which degradation occurs cannot be defined from the averaged thermograms’ weight and temperature (Figure 2a). The derivative curve is, therefore, presented in Figure 2b. 

Table 2 and Figure 2b show that the *T_max_* of the PLA blends slightly increased, and the lost weight rate was reduced when the waste content increased. The reason could be the presence of polymers with different thermal histories in the structure of the PLA blends. Additionally, PLA and its blends degraded in a narrow temperature range of about 30 °C. The results of the TGA, therefore, indicated that adding PLA debris to the virgin material did not significantly impact the thermal stability of the 3D-printed sample.

### 3.2. Differential Scanning Calorimetry

All of the figures in Table 3 and Figure 3 present the results of the DSC test. Analysing the evolution of the thermal properties with increasing recycled content, it is observed that the *T_g_* of the PLA blends slightly varied between 58 and 61 °C for the first heating and between 60 and 64 °C for the second heating. Likewise, the *T_m_* fluctuated between 173 and 177 °C, which can be attributed to the melting peak of α crystals, normally observed around 180 °C [47]. In addition, as the *T_m_* depends on the flexibility of the polymer chain, no greater mobility of the macromolecular chain could be supposed [12].

Regarding the measured enthalpies, reductions were observed in both the Δ*H_c_* and Δ*H_m_* with increasing recycled content. However, the Δ*H_c_* was not revealed during the second heating because the cooling rate for PLA 3D-printed specimens at 2 °C/min was too slow. The magnitude of the crystallization exotherm decreased when the sample was more slowly cooled, and it vanished at scan rates equal to or lower than 10 °C/min [48]. The cooling of the 3D-printed specimens at an ambient temperature had, therefore, slow crystallization kinetics.

The V25R75 and V50R50 DSC thermograms in Figure 3b, after the second heating, had two melting peaks. A double melting peak is a common phenomenon for polymers such as poly (ethylene terephthalate), isotactic polystyrene, poly (ether ether ketone), and poly (ether imide). The reason could be the presence of two different crystal or morphological structures in the initial sample, but it is generally the result of annealing during the DSC scans whereby crystals of a low perfection melt have time to recrystallize a few degrees above and to remelt [48]. When the scan rate is low, i.e., 10 °C/min, there is enough time for the thinner crystals to melt and recrystallize before giving a second endotherm at a higher temperature [48].

According to [12], the degree of crystallinity, *X_c_*, is defined as follows:(1)$${X_{c}}_{\;}=\frac{\Delta H_{m}}{\Delta H^{*}}\times 100$$
where Δ*H_m_* is the heat of melting, and Δ*H** denotes the heat of melting for an infinitely large crystal. Some authors have used the value of 93 J/g [12] or 106 J/g [49]. In this work, 106 J/g was used. Table 3 shows that the degree of crystallinity was reduced from 48% for neat PLA 3D-printed specimens to 30% for specimens with 75% of waste. These values had a linear tendency, with a regression of *R*^2^ = 0.92, thus reducing the crystallinity by 0.26% per percentage of the recycled PLA added. As the crystallization process depends on the molecular weight of PLA [49,50,51,52,53], the PLA waste used in this experiment had low crystallization kinetics. 

It is worth stressing that *T_m_*, *T_c_*, and *T_g_* of V25R75 were almost the same for the pure PLA specimens. However, the V25R75 PLA blend had a more amorphous nature because the crystallization exotherm and melting endotherm had identical energy content (the same area). The fracture mechanism and the behaviour of the 3D-printed specimens were, therefore, influenced during the tensile test. Nevertheless, the DSC curves of the PLA blends had several degradation peaks, thus confirming the presence of additives in the PLA filament that the manufacturers use to improve the PLA’s quality.

Despite the fact that the filament from 100% of PLA waste was not suitable for 3D printing, the DSC results show that the *T_g_* and *T_m_* are 59.36 °C and 174.99 °C after the second heating, respectively. These characteristics are in the same numerical range as the *T_g_* and *T_m_* of V100R0, V75R25, V50R50, and V25R75. Decreasing tendency for both the Δ*H_c_* and Δ*H_m_* is preserved. Taking into account that PLA degrades during thermal processing, rapidly reducing the molecular weight [54], it can be predicted that the *T_g_*, *T_m_*, Δ*H_c_*, and Δ*H_m_* of the 3D-printed samples from 100% waste could be lower.

### 3.3. Scanning Electron Microscopy

The fracture surfaces of the V100R0, V75R25, V50R50, and V25R75 PLA blend specimens were studied in the post-tensile test condition through SEM to characterize the fracture surface. The SEM images of the samples are shown in Figure 4 and Figure 5. Figure 4 shows that the fracture surface (the cross-section view) significantly changed from one sample to another. First, the 3D-printed beads were visible in the pure PLA and were gradually softened as the recycled content increased, thus showing the V75R25 as a more continuous matter. Second, the roughness also changed: V100R0 depicted a brittle fracture crack with clean and sudden surfaces, while in the remaining samples, there was a rougher surface with features of a more ductile rupture where the material underwent slight plastic deformation. 

Figure 5 presents the micrographs acquired at a higher magnification to analyse and determine structural changes in the blends. All of the micrographs showed materials with similar characteristics: a porous matrix with homogeneous SEM secondary electron contrasts, which means that the PLA blends were not exposed to any other material content when preparing the FFF filaments, and some strands of fibres protruded from inside of some of the holes. The strands seemed to be greater when the recycled content increased. The holes between these fibres and their surrounding matrix are shown. The micro-pores’ density, which was measured from high magnification SEM images such as those in Figure 5, was gradually reduced from 6.4·10^9^ cm^−2^ in V75R25 to 2.5·10^9^ cm^−2^ in V25R75. 

### 3.4. Mechanical Properties

Figure 6 shows the results of the tensile tests. The maximum strength values of all of the samples were within 95% confidence intervals of the values counted, according to ISO 2602 [55]. The tensile strength increased with the increasing percentage of recycled material, from 44.2 ± 2.18 MPa for pure PLA to 52.61 ± 2.28 MPa for 75% of recycled PLA loading. 

Figure 6a shows the tensile stress–strain graphs of the 3D-printed specimens from the PLA blends with different levels of PLA waste. Only one of the five specimens per blend, which represented the overall behaviour, is shown here for comparison. According to ISO 527 [44], the curves of V100R0, V75R25, and V50R50 corresponded to brittle materials with no yield point. Samples from the V25R75 PLA blend showed a much more ductile behaviour, with a ductility higher than 13% (measured as the percentage of elongation), with 3% being the average value for the other three PLA blends. Moreover, the inset in Figure 6a shows that the specimens from the PLA blends without waste and with 25 and 50% of post-consumer polymer experienced a brittle fracture. In these cases, the neck was not formed, so there was neither fluidity nor hardening of the material. 

The mechanical properties of polylactides beyond the elastic region are dependent on the amount and type of crystalline regions developed during processing [47,53]. In addition, a greater degree of crystallization in semicrystalline thermoplastics means a lower content of the free volume, and the stiffness generally increases [47]. The DSC test revealed that the V25R75 PLA blend was more amorphous than the other mixtures. Hence, this could be the reason for the strong increase in ductility. Additionally, the slight reduction in the *T_g_* when the PLA recycled content increased could also influence this fact.

The rising trend of tensile strength with increasing recycled content can be explained by the micrographs of the SEM depicted in Figure 4. In Figure 4a, bigger voids between the 3D-printed layers of the V100R0 sample are clearly seen than they are for V25R75, V50R50, and V75R25. Therefore, the higher value of tensile strength could be due to better inter-layer bonding for the samples with the recycled PLA. It is a well-known fact that the melt flow index increases for recycled PLA [28,56], which means that recycled PLA filament tends to flow better. Therefore, it can be supposed that the tensile strength continues to increase with a larger content of waste because the number and size of the voids decrease.

## 4. Conclusions

In this study, we evaluated the recyclability of PLA FFF-printed parts. Hence, FFF filaments from recycled PLA feedstock from real waste mixed with virgin PLA pellets in three different proportions (25%, 50%, and 75%) were produced with a constant cross-section and good flowability. 

Based on the results of the thermal, morphological, and mechanical tests of 3D-printed specimens, the following conclusions have been drawn:The crystallinity degree dropped when the percentage of post-consumer PLA increased from 48% in V100R0 to 30% in V25R75. The reason could be the fact that the molecular chains of the secondary PLA were too short to organize the crystals;SEM micrographs of the fracture surface showed that virgin PLA specimens were more brittle and less dense than recycled PLA blends, thus significantly reducing both the millimetre- and micrometre-sized holes;The mechanical test showed that 3D printing with recycled PLA was a viable option; the tensile strength increased with the recycled content by 19% compared with the PLA samples.

It must be mentioned that all three blends showed good FFF processability, and there was not any clogging during printing. It is worth stressing that there was a detriment in the processability during the production of the filaments obtained from 100% recycled sources. Another problem during the conduction of this experiment was that the filament from the blend with the 75% content of PLA waste was brittle, although its printed samples had the highest tensile strength. However, this study indicates that the mechanical properties of the reprocessed parts and their basic association are better than those made up of virgin material. Further studies should, therefore, focus on solving the filament embrittlement problem by finding suitable additives or proposing a different AM method, such as fused granulated fabrication, where the material is applied directly in the form of granules.

## Figures and Tables

**Figure 1 polymers-15-02165-f001:**
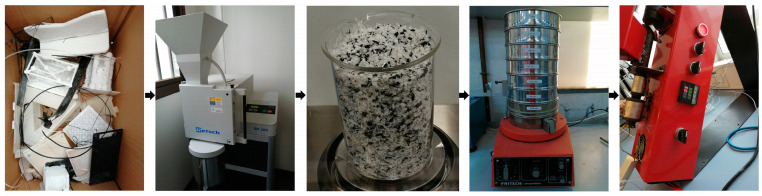
Preparing the filament from PLA debris. From left to right: PLA debris collection, shredding, shredded PLA, sieving, and filament production.

**Figure 2 polymers-15-02165-f002:**
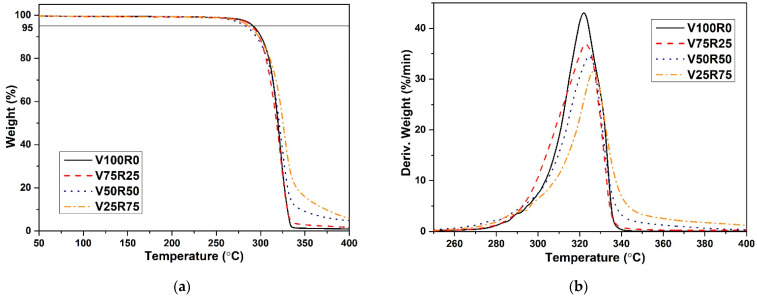
TGA curves of the PLA blends with different waste content. (**a**) The evolution of the weight versus the temperature; (**b**) The derivative curve.

**Figure 3 polymers-15-02165-f003:**
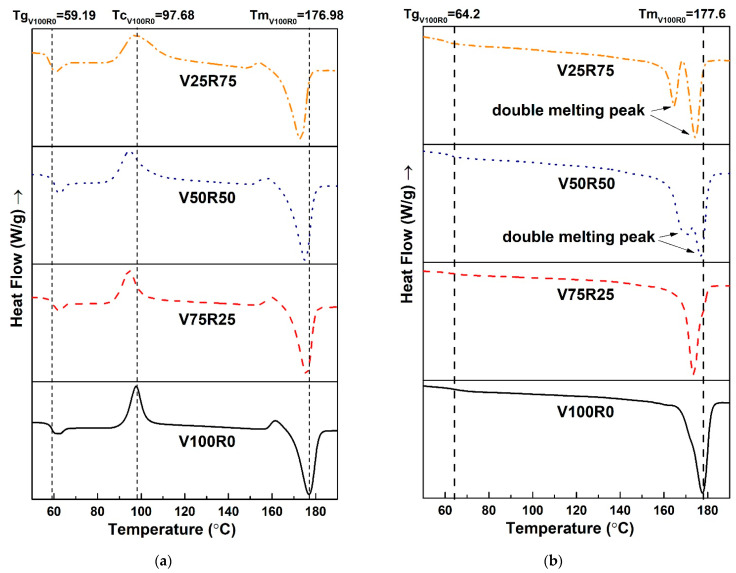
DSC curves of the PLA blends with various waste content at first heating (**a**) and second heating (**b**) after a 2 °C/min cooling step.

**Figure 4 polymers-15-02165-f004:**
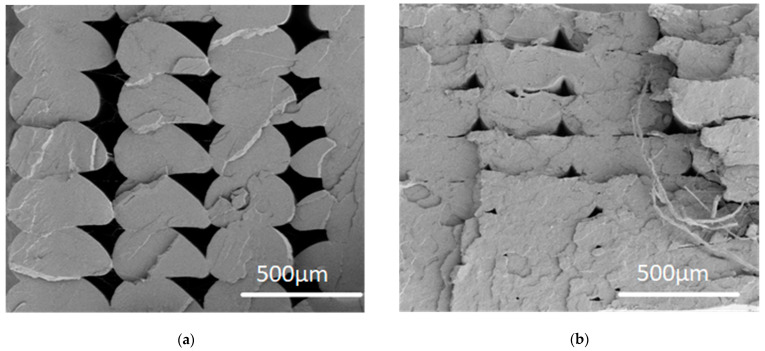
SEM secondary electron micrographs of the fracture surface of the PLA blends: (**a**) V100R0, (**b**) V75R25, (**c**) V50R50, and (**d**) V25R75. Scale bar: 500 μm.

**Figure 5 polymers-15-02165-f005:**
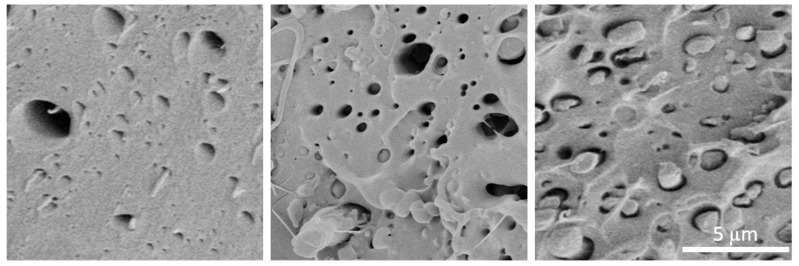
SEM images from 12 × 12 µm fracture regions of samples V75R25 (**left**), V50R50 (**middle**), and V25R75 (**right**).

**Figure 6 polymers-15-02165-f006:**
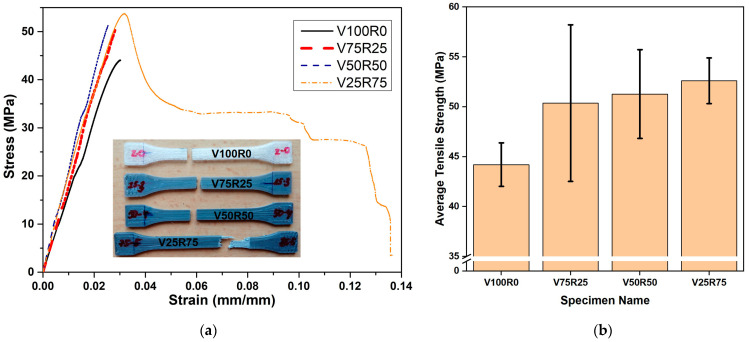
(**a**) Tensile stress–strain curves of the PLA blends. The inset picture shows the samples after the tensile test. (**b**) Resultant tensile strengths.

**Table 1 polymers-15-02165-t001:** Blend of compositions and sample codes.

Sample Code	Weight Ratio (%)	
	Virgin PLA	Recycled PLA
V100R0	100	0
V75R25	75	25
V50R50	50	50
V25R75	25	75

**Table 2 polymers-15-02165-t002:** Thermogravimetric analysis results of the samples: the temperature of the 5% wt. mass loss (*T*_*loss*5%_), initial and final degradation temperatures and the difference between them, and the maximum temperature of polymer degradation (*T_i_*, *T_f_*, (*T_f_-T_i_*), *T_max_*, respectively) computed according to ISO 11358-1:2014 [43] are shown. All values are given in °C.

Sample Code	*T_5%loss_*	*T_i_*	*T_f_*	*T_f_-T_i_*	*T_max_*
V100R0	292	306	334	28	322
V75R25	290	304	332	28	323
V50R50	283	305	335	30	325
V25R75	286	308	340	32	327

**Table 3 polymers-15-02165-t003:** Results of DSC tests: glass transition, crystallization, and melting temperatures (*T_g_*, *T_c_*, and *T_m_*, respectively) are shown, as well as the enthalpy of crystallization and fusion (Δ*H_c_* and Δ*H_m_*) and the calculated degree of crystallinity (*X_c_*).

Sample Code	First Heating					Second Heating			*X_c_*
	*T_g_*	*T_c_*	Δ*H_c_*	*T_m_*	Δ*H_m_*	*T_g_*	*T_m_*	Δ*H_m_*	
	°C	°C	J/g	°C	J/g	°C	°C	J/g	%
V100R0	59.19	97.68	45.41	176.98	57.24	64.20	177.60	50.86	48
V75R25	59.84	94.45	37.34	175.80	50.25	63.20	173.50	49.91	47
V50R50	60.71	94.63	27.38	175.01	36.21	60.98	176.82	38.24	36
V25R75	58.43	97.87	24.85	173.25	28.51	60.12	174.31	31.62	30

## Data Availability

Data are available upon reasonable request by contacting the corresponding author.

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
