# Peer review of "Manufacture and Characterization of Polylactic Acid Filaments Recycled from Real Waste for 3D Printing"

_polymers, 2023, doi:10.3390/polym15092165_

Round 1

Reviewer 1 Report

The authors have investigated the thermal, morphological, and mechanical properties of 3D-printed PLA blends of virgin and recycled material.

Here are some points that need to be addressed before publication of the paper.

·       What is the novelty of the work? Clarify this in the last paragraph of the introduction. The proportion used cannot be the novelty of a research work unless you state why these proportions.

·       Add more references regarding filament making from waste materials, polymers, and fibres, not just PLA, to show the effective parameters in filament making process that needs to be optimized such as the below ones:

[a] Short basalt fibre reinforced recycled polypropylene filaments for 3D printing

[b] A review on PLA with different fillers used as a filament in 3D printing

[c] "Recovery of Particle Reinforced Composite 3D Printing Filament from Recycled Industrial Polypropylene and Glass Fibre Waste," Proceedings of the World Congress on Mechanical, Chemical, and Material Engineering, 2022.

·       Page 2, line 94: add a reference here for the properties of the polymer.

·       Add more details regarding the optimized filament-making process parameters such as speed, and heat zone temperatures in the Noztek extruder.

·       Introduce the temperatures presented in Table 2.

·       Where you use Tg for the first time, you must introduce that in its full form name, glass transition temperature. The same for Tm and delta-hs.

·       Why did the samples with higher waste PLA proportions show higher tensile strength? Explain.

·       What about the samples printed from 100% recycled PLA?

·       Summarise the main achievements of the work by bullets in the conclusion section.

Reviewer 2 Report

The manuscript 237024 reports the preparation of the 3D printed blend of virgin and recycled PLA filament and a series of experimental techniques conducted to characterize these blends. The authors did a lot of work, and the full text is organized logically, comprehensively, and fairly well. The following comments are in detail for the authors to consider.

1.       The Materials and Methods should divide into sub-sections like 2.1. Materials, 2.2. 3D printing filament preparation.

2.       Line 205, Ref. 7 doesn’t discuss PLA crystallization and it might be a wrong ref. Please check other references.

3.       It would be better if it can include a DSC and TGA thermograph of a raw filament of recycled PLA (although the filament was curly and not printable) 
